# Comprehensive Analysis of the Role of Fibrinogen and Thrombin in Clot Formation and Structure for Plasma and Purified Fibrinogen

**DOI:** 10.3390/biom14020230

**Published:** 2024-02-16

**Authors:** Rebecca A. Risman, Heather A. Belcher, Ranjini K. Ramanujam, John W. Weisel, Nathan E. Hudson, Valerie Tutwiler

**Affiliations:** 1Department of Biomedical Engineering, Rutgers University, Piscataway, NJ 08854, USA; rr901@scarletmail.rutgers.edu (R.A.R.); rk931@scarletmail.rutgers.edu (R.K.R.); 2Department of Physics, East Carolina University, Greenville, NC 27858, USA; belcherh17@students.ecu.edu (H.A.B.); hudsonn16@ecu.edu (N.E.H.); 3Department of Cell and Developmental Biology, University of Pennsylvania, Philadelphia, PA 19104, USA; weisel@pennmedicine.upenn.edu

**Keywords:** fibrin, fibrinogen, thrombin, tissue factor, thrombosis

## Abstract

Altered properties of fibrin clots have been associated with bleeding and thrombotic disorders, including hemophilia or trauma and heart attack or stroke. Clotting factors, such as thrombin and tissue factor, or blood plasma proteins, such as fibrinogen, play critical roles in fibrin network polymerization. The concentrations and combinations of these proteins affect the structure and stability of clots, which can lead to downstream complications. The present work includes clots made from plasma and purified fibrinogen and shows how varying fibrinogen and activation factor concentrations affect the fibrin properties under both conditions. We used a combination of scanning electron microscopy, confocal microscopy, and turbidimetry to analyze clot/fiber structure and polymerization. We quantified the structural and polymerization features and found similar trends with increasing/decreasing fibrinogen and thrombin concentrations for both purified fibrinogen and plasma clots. Using our compiled results, we were able to generate multiple linear regressions that predict structural and polymerization features using various fibrinogen and clotting agent concentrations. This study provides an analysis of structural and polymerization features of clots made with purified fibrinogen or plasma at various fibrinogen and clotting agent concentrations. Our results could be utilized to aid in interpreting results, designing future experiments, or developing relevant mathematical models.

## 1. Introduction

During hemostasis, a blood clot forms to stop blood flow from exiting the wound site [1]. However, unnecessary blood clotting in the veins or arteries can form an obstructive thrombus that blocks blood flow to downstream organs and can ultimately lead to an ischemic stroke, heart attack, or venous thromboembolism (VTE) [2]. Following vessel injury, levels of procoagulant factors are upregulated in the blood, which can ultimately lead to fibrin polymerization and clot formation [3,4]. Fibrinogen, the precursor to fibrin, is a 340 kDa glycoprotein consisting of two subunits, each containing three polypeptide chains: Aα, Bβ, and γ chains [5,6]. These subunits are linked together by disulfide bonds in the central E region. From both sides of the E region, the three chains extend to the two distal D regions. The Bβ and γ chains end in two compact globular nodules, the β and γ nodules, which contain hole ‘a’ and hole ‘b’, and the Aα chain extends into a partially unstructured part of the molecule termed the αC region [7]. After vessel injury, active thrombin cleaves fibrinopeptides A and B to make monomeric fibrin, which polymerizes into half-staggered protofibrils that aggregate laterally and form a three-dimensional network [8,9]. This fibrin network provides the structural and mechanical stability that is essential for blood coagulation and wound healing [5,10]. However, a clot can stem blood flow to stop bleeding but can also occlude a blood vessel and restrict the transportation of essential nutrients to downstream organs [11,12,13].

Dangerous thrombotic events, such as stroke, heart attack, or VTE, have been associated with altered structural properties of the in vitro plasma clots made from the blood of these patients [14,15,16,17,18,19]. Diseases such as cancer [20,21], diabetes [14,22], and COVID-19 [23,24] have also been linked to the altered structural properties of in vitro plasma clots, which could potentially impede clot degradation and increase the risk of thrombotic events [25]. The coagulation cascade is a complex process; irregularities at any step can lead to aberrant clot structures [26]. To understand why some diseases are linked with altered fibrin structure, it is first necessary to understand what factors cause structural changes to fibrin networks. To interpret fibrin clot behavior in physiological conditions, clots can be made using plasma. However, plasma contains many proteins (including fibrinogen) that can bind to fibrin clots, changing the clot structure [27]. For example, albumin, gamma-globulin, antithrombin III, and fibronectin all alter fibrin fiber thickness as well as clot polymerization kinetics [4,28]. The fibrin network structure has also been studied using purified fibrinogen, allowing for analysis of the effects of thrombin or fibrinogen concentration on individual parameters (i.e., density, diameter, pore size) in the absence of the other plasma proteins and their confounding influences [29,30,31]. While experiments have been performed to study clot structure, there is a need for a comprehensive analysis using a combination of standard techniques [32]. In this work, we carry out a systematic study on the effect of varying fibrinogen and activation factor concentrations on the fibrin network structure, utilizing a combination of modeling and laboratory experimental techniques to investigate clots made from both plasma and purified fibrinogen.

## 2. Materials and Methods

A combination of multiple laboratory experimental approaches and modeling was used to compare fibrin network polymerization and structural parameters across varying fibrinogen (i.e., 1–10 mg/mL) and activation factor concentrations (i.e., thrombin: 0.1–1 U/mL). Detailed methods are available in the Appendix A.

Turbidity assays were used to compare polymerization of the clots. These assays exploit the increase in light scattering as fibrin polymerizes. They can be used to determine the lag time, rate of polymerization, and maximum absorbance and/or turbidity, which is related to the fiber density and diameter; however, turbidity assays require consistent mixing/pipetting and measurement times to be compared between samples [33]. Scanning electron microscopy (SEM) and confocal microscopy were used to both quantitatively and qualitatively compare clot structural properties. Confocal microscopy can provide two- and three-dimensional organizational information about the clot geometry, such as pore size and density. However, confocal results are diffraction limited and unable to distinguish features below ~200 nm [34], making them incapable of reliably determining fiber diameter. SEM has higher resolution than confocal microscopy, which allows for the measurement of individual fiber diameters. However, SEM preparation may result in morphological changes within the clot due to drying, fixing, dehydration, and sputter coating. Super-resolution microscopy is likely the most accurate method to measure fibrin fiber diameters but is not widely available. Although we did not utilize super-resolution microscopy in this work, it has been found that SEM and super-resolution microscopy result in very similar diameters across the range of physiologically relevant fibrinogen concentrations [35]. These results suggest that there are not significant structural changes to the fibers due to the SEM sample preparation process or that any shrinkage is compensated for by metal deposited during sputter coating. Lastly, turbidimetry, the process of taking turbidity measurements at several wavelengths, was used as another technique to determine individual fiber diameters. However, turbidimetry only allows for the determination of the average diameter and can therefore be skewed by very large or small fibers, and it does not account for multiple scattering [36,37].

### 2.1. Purified Fibrinogen vs. Plasma

Purified fibrinogen experiments were performed at East Carolina University (ECU) while plasma experiments were performed at Rutgers University (RU). Purified fibrinogen was purchased from Enzyme Research Laboratories (Peak 1 Fib, XIII free, P1 FIB). Commercially available human-pooled plasma from more than 25 healthy donors was purchased from Cone Bioproducts (#5781). Individual replicates were conducted with the same lot of pooled plasma. Turbidity approaches using ECU reagents (thrombin and FXIII) were reproducible at RU (Appendix A). Four concentrations of the experimental parameter (fibrinogen or thrombin/tissue factor (TF)) were used for most experiments (with the exception of samples with a fixed ratio of tissue factor to fibrinogen, in which only two concentrations were used).

### 2.2. Scanning Electron Microscopy (SEM) Imaging

Clots made from plasma and purified fibrinogen were allowed to stand for at least 30 min with the conditions described in detail in the Appendix A. Plasma clots were formed at room temperature [38]; clots from purified fibrinogen were formed at 37 °C. Structural comparisons of room temperature (RT) vs. 37 °C clots can be seen in the Appendix A; any changes were smaller in magnitude than differences resulting from the effects of changing concentration. Clots were washed with buffer, fixed with glutaraldehyde, and dehydrated with an ethanol series and dried with hexamethyldisilazane (HMDS). Samples were sputter coated and imaged using university-respective (ECU’s Zeiss EVO 10 or RU’s Zeiss SIGMA field emission) scanning electron microscopes. At least three images per sample replicate were acquired with a minimum of two replicates and six images. Fibrin fiber diameters (minimum 50 fibers per image) were measured using Fiji (Fiji Is Just ImageJ) Version 2.9.0/1.53t.

### 2.3. Confocal Microscopy

To prepare the clot samples for confocal microscopy, fibrinogen, thrombin/TF, calcium, and labeled fibrinogen were allowed to fully form for at least 30 min. Plasma clots were formed in a 96-well plate at room temperature (as described in Risman et al., 2022) [39]; clots from purified fibrinogen were formed on a cover glass at 37 °C. The same sample preparation was used for samples containing each of the fibrinogen/thrombin concentrations analyzed (described in the Appendix A). At least two images per sample replicate were acquired using a confocal microscope (average of 6 images, ECU’s Zeiss LSM800 or RU’s Zeiss LSM800). The percent area covered by fibrin fibers, the pore size of fibrin networks, and the length of fibrin fibers were measured using FIJI of images in a single plane of the clot. The percent area covered by the fibers represents the fluorescent density (the percentage of the 2D images that contained fluorescence after being made binary). Pore size measurements were taken using a line tool of the average diameter of the pore on 2D slices (single plane) of the clots. While technically the pore sizes are three-dimensional, it has been shown that 2D pore size measurements can accurately represent the three-dimensional pores [40,41].

### 2.4. Turbidity/Turbidimetry

The absorbance measurements of forming clots (with plasma or fibrinogen and TF or thrombin) were taken every 15 s for 1 hour at a wavelength of 405 nm and a temperature of 37 °C to capture the polymerization process. Three replicates were measured. After that hour, absorbance measurements were taken on the same wells every 10 nm from 500 to 800 nm as well as at a wavelength of 900 nm and 977 nm (in order to calculate the path length). To calculate the turbidity values (τ) at each timepoint for the samples, the absorbance values with the background subtracted out (*A*) were divided by the path length (*x*) and then multiplied by *ln*(10) such that:$$\tau =\frac{A}{x}*ln(10)$$

Three parameters were analyzed from the turbidity assays: lag phase, maximum turbidity, and rate of clot formation.

There were multiple turbidimetry approaches used to determine fibrin diameter from light-scattering measurements, as there are several different approaches commonly used within the fibrin community. We utilized the approaches introduced by Carr and Hermans [42], and the corrected approach introduced by Yeromonahos et. al. [43], as these approaches have been found to be reasonably accurate for clots made from purified fibrinogen under clotting conditions similar to those utilized in this study when compared to super-resolution microscopy [44]. The methods of determining the fiber diameter from the turbidimetry data using these two approaches are described in the Appendix A [45,46,47].

To compare the microplate readers (ECU’s BioTek Synergy HT and RU’s SpectraMax) and overall techniques utilized at ECU and RU for turbidity and turbidimetry approaches, ECU reagents (FXIII and thrombin) were utilized at RU. Experiments were conducted using RU protocol. Appendix A shows the overlap of the data indicating the similarity of the technique and equipment.

### 2.5. Multiple Linear Regressions

To delineate the dependent role of fibrinogen and thrombin concentrations, we developed multiple linear regressions for each independent parameter (diameter, pore size, % area, fiber length, lag time, rate of formation, and maximum turbidity). A custom python code with inputs of average independent variable data was used to output multiple linear regression equations and r-squared values. These codes can be found on a GitHub repository at https://github.com/rr901/Fibrin-structure (accessed on 8 February 2024).

### 2.6. Statistical Analysis

All statistical analysis was performed using GraphPad Prism 10.0.2. Outliers were identified and removed using the Grubbs’ test with an alpha of 0.05. To perform comparison tests on the diameters, pore size, percent area covered, fiber length, and turbidity measurements, normality was first confirmed with the D’Agostino and Pearson test. One-way ANOVA tests were used to compare concentrations and techniques; corrected ANOVA tests were used when samples were non-normal (Kruskal–Wallis test), and Welch’s corrected ANOVA was used for samples with different standard deviations. Corresponding post hoc tests were performed to conduct multiple comparisons. The specific test used is specified in each figure legend. Turbidimetry-derived values were compared by uncertainty analysis. Simple linear regressions were calculated using Prism by considering the mean Y value for each data point. R-squared and slopes were recorded.

Bar plots display the mean ± standard error of the mean for parameters with normal distributions; median ± interquartile range for parameters with non-normal distributions. XY graphs are displayed as the mean ± standard error of the mean with connecting lines. Unless otherwise specified, the following star nomenclature is used for statistical significance: ns (not significant) *p* > 0.05, * *p* < 0.05, ** *p* < 0.01, *** *p* < 0.001, **** *p* < 0.0001.

## 3. Results

A variety of modeling and experimental techniques were used to analyze fibrin polymerization and network structure. We used a combination of SEM imaging, confocal imaging, and turbidity/turbidimetry to study clots made from purified fibrinogen and pooled human plasma for a range of physiologically relevant fibrinogen and thrombin concentrations. These experimental techniques were chosen as they are among the most commonly utilized for studying fibrin polymerization and structure. Pooled human plasma was used to minimize the large variations between individual plasmas.

### 3.1. Effect of Fibrinogen Concentration on Fibrin Network Structure

We assessed the effect of fibrinogen concentration (1, 2.7, 5, and 10 mg/mL) on the resulting fiber and network structure in clots made from purified fibrinogen (Figure 1) and pooled human plasma (Figure 2) with a fixed thrombin concentration of 0.1 NIH-U/mL for the purified fibrinogen samples and 75 pM tissue factor for plasma samples, which is the concentration of TF used for diagnostic testing [48,49]. For both sample types, fiber diameter increased with increasing fibrinogen concentration (Figure 1A–D,I and Figure 2A–D,I). The plasma samples increased by a rate of 29 nm per mg/mL (R^2^ = 0.90), while the purified fibrinogen samples increased by a rate of 11 nm per mg/mL (R^2^ = 0.96). Furthermore, both purified fibrinogen and plasma clots had decreased pore sizes with increasing fibrinogen concentration (purified fibrinogen: 0.72 μm per mg/mL; plasma: 0.24 μm per mg/mL rate of decrease (R^2^ = 0.83 and 0.70, respectively)) (Figure 1E–H,J and Figure 2E–H,J). They also both had increased fiber area covered with increased fibrinogen concentration (purified fibrinogen: 3.50% area per mg/mL rate of increase (R^2^ = 0.71), plasma: 0.96% area per mg/mL increase (R^2^ = 0.53)) (Figure 1E–H,K and Figure 2E–H,K). Both samples had decreased fiber length with increasing fibrinogen concentration (purified fibrinogen: 0.64 μm per mg/mL (R^2^ = 0.81), plasma: 0.63 μm per mg/mL (R^2^ = 0.79)) (Figure 1E–H,L and Figure 2E–H,L).

For both purified fibrinogen and plasma, according to both turbidimetric fitting approaches and according to SEM imaging, there was an increase in diameter as the fibrinogen concentration increased (Figure 3). For both samples, the Carr–Hermans approach usually provides the largest diameter value, and the Yeromonahos approach usually provided the smallest diameter with the SEM diameter falling somewhere between the two. Diameters of fibrin fibers of clots made with plasma measured with SEM, Carr–Hermans, and Yeromonahos decreased with slopes of 28.7 (R^2^ = 0.9044), 51.3 (R^2^ = 0.9459), and 4.1 (R^2^ = 0.9373) nm per mg/mL, respectively. Diameters of fibrin fibers made with purified fibrinogen measured with SEM, Carr–Hermans, and Yeromonahos increased with slopes of 10.97 (R^2^ = 0.9598), 24.8 (R^2^ = 0.9922), and 4.0 (R^2^ = 0.9763) nm per mg/mL, respectively. This showed that while the trends remained, the diameter differences between fibrinogen concentrations were muted for the Yeromonahos approach and exaggerated for the Carr–Hermans approach compared to SEM. The diameter values obtained from the three different methods were all similar for fibrinogen concentrations of 1 and 2.7 mg/mL, but as the fibrinogen concentration increased, so did the variation in the diameter values obtained by the different methods with a very large increase in the diameter obtained using the Carr–Hermans approach at 10 mg/mL for both samples. The raw turbidimetry curves are shown in Appendix A.

To further isolate the role of fibrinogen concentration, we preserved the volume ratio of tissue factor to fibrinogen (3.1 mM TF for 1 mg/mL of fibrinogen). We did this by decreasing the concentration of fibrinogen, just as in the previous plasma samples; however, we accordingly decreased the tissue factor concentration to provide a fixed ratio between the two. In the above plasma samples, there was a fixed volume ratio (1:80), and thus there was a smaller ratio of fibrinogen molecules to TF molecules as the fibrinogen concentration increased, while the TF concentration remained constant. Here, we focused on two fibrinogen concentrations: 1 and 2.7 mg/mL. The latter sample (2.7 mg/mL) is the same as in the previous plasma samples, while the former (1 mg/mL) has less TF. For this reason, we are able to distinguish the role of fibrinogen independent of tissue factor. The results are shown in Appendix A. As fibrinogen concentration increased from 1 to 2.7 mg/mL, we observed thicker fiber diameter (Appendix A, *p* < 0.001), reduced pore size (Appendix A, *p* < 0.001), denser networks (Appendix A, *p* < 0.01), and longer fibers (Appendix A
*p* < 0.001). We were also able to compare diameters as measured with SEM when there was a fixed volume ratio of TF to fibrinogen. Interestingly, with a lower fibrinogen concentration and less TF, the diameter was thicker (Appendix A). Compared to SEM, the Carr–Hermans turbidimetry fitting approach underestimated the diameter of the 1 mg/mL sample and overestimated the diameter of the 2.7 mg/mL sample. Yeromonahos only slightly underestimated the diameter compared to SEM but followed a more similar trend.

### 3.2. Effect of Fibrinogen Concentration on Fibrin Network Formation

We quantified fibrin polymerization properties using turbidity. For both clots made from purified fibrinogen and plasma, the maximum turbidity values and rate of polymerization increased with increasing fibrinogen concentration (Table 1, Appendix A). Maximum turbidity is often used as a quantitative representation of the clot density and fiber thickness. We found that the maximum turbidity increased as the % area of fibers increased and as the fiber diameter increased (Appendix A). The rates of clot formation increased with increasing fibrinogen for the plasma samples and purified fibrinogen samples. The lag time increased as the fibrinogen concentration increased for the clots made with purified fibrinogen, but it decreased for the clots made with plasma. The raw turbidity curves are shown in Appendix A. The relationship between maximum turbidity and fibrinogen/thrombin concentration (Appendix A), % area (Appendix A), and diameter (Appendix A) can be seen in the Appendix A.

### 3.3. Effect of Thrombin Concentration on Fibrin Network Structure

We then assessed the effect of thrombin concentration (0.1, 0.25, 0.5, and 1 U/mL) on the resulting fiber and network structure in clots made from purified fibrinogen (Figure 4) and pooled human plasma (Figure 5). For both sample types (plasma and purified fibrinogen), the diameter decreased with increasing thrombin concentration (Figure 4A–D,I and Figure 5A–D,I) (plasma samples: 100 nm per U/mL (R^2^ = 0.82), purified fibrinogen: 91 nm per U/mL (R^2^ = 0.94)). Both samples also have decreased pore size and fiber length as the thrombin concentration increases (Figure 4E–H,J,L and Figure 5E–H,J,L) (pore size rate of decrease 3.62 μm per U/mL (R^2^ = 0.89) for purified fibrinogen and 3.58 μm per U/mL (R^2^ = 0.61) for plasma; fiber length rate of decrease 4.08 μm per U/mL (R^2^ = 0.98) for purified fibrinogen and 5.67 μm per U/mL (R^2^ = 0.67) for plasma). The density had an overall increasing trend for both purified fibrinogen and plasma samples as the thrombin concentration increased with a rate of increase of 8.42% area per U/mL (R^2^ = 0.60) for purified fibrinogen and 12.87% area per U/mL (R^2^ = 0.83) for plasma (Figure 4E–H,K and Figure 5E–H,K).

Figure 6 shows the diameters obtained from SEM in comparison to those obtained from the Carr–Hermans and Yeromonahos approaches fit to turbidimetry data for both purified fibrinogen and plasma at the four different thrombin concentrations. All three methods, for both samples, showed a decrease in diameter as the thrombin concentration was increased. The diameters of fibrin fibers in clots made with plasma measured with SEM, Carr–Hermans, and Yeromonahos decreased with slopes of −100.1 (R^2^ = 0.81), −81.0 (R^2^ = 0.79), and −10.8 (R^2^ = 0.87), respectively. Diameters of fibrin fibers in clots made with purified fibrinogen measured with SEM, Carr–Hermans, and Yeromonahos decreased with slopes of −90.4 (R^2^ = 0.94), −66.8 (R^2^ = 0.89), and −25.9 (R^2^ = 0.95), respectively. Like what was seen for increasing fibrinogen, the Yeromonahos approach had muted trends compared to Carr–Hermans and SEM. For purified fibrinogen, the Carr–Hermans and SEM diameters were very similar for 0.1 U/mL and the Yeromonahos and SEM diameters were very similar for 0.25 and 0.5 U/mL, but the diameter obtained from SEM imaging was much lower than the other two methods for 1 U/mL. For the plasma samples, there was a large difference in the diameter values obtained using the three different methods, with the SEM values being the smallest, the Yeromonahos values being the next largest, and the Carr–Hermans diameter values being the largest. The raw turbidimetry curves are provided in Appendix A.

### 3.4. Effect of Thrombin Concentration on Fibrin Network Formation

The results obtained from the turbidity curves for the clots made with purified fibrinogen and plasma made at each of the thrombin concentrations are provided in Table 2. The raw turbidity curves are shown in Appendix A. For both the purified fibrinogen and for the plasma samples, the lag time decreased as the thrombin concentration increased. For the purified fibrinogen samples, the rate of polymerization increased as the thrombin concentration increased, while it decreased for the plasma samples. The maximum turbidity values decreased slightly as the thrombin concentration increased for the purified fibrinogen samples but increased slightly overall for the plasma samples. Interestingly, the maximum turbidity decreased with increasing % area for purified fibrinogen but increased with increasing % area for plasma (Appendix A). Similar to what was seen in Section 3.2, the maximum turbidity increased with increasing diameter when the clots were made with purified fibrinogen but decreased with increasing diameter for clots made with plasma (Appendix A).

### 3.5. Generalized Trends and Predicted Values

First, we looked at trends developed when the clots are dependent on fibrinogen and thrombin concentrations, as summarized in Appendix A. Most parameters, both structural and kinetic, show the same increasing or decreasing trends for clots made with purified fibrinogen or plasma. For clots made with increasing fibrinogen concentration, the diameter increased for all methods, the percent area and maximum turbidity increased, the rate of formation was faster, and the pore size and fiber length decreased. For clots made with increasing thrombin, the diameter decreased for all methods, the pore size and fiber length decreased, the percent area fraction increased, and the lag time shortened. For clots made with purified fibrinogen, the lag time and fibrinogen concentration were positively related, whereas for plasma clots, the fibrinogen concentration and lag time were inversely related. For clots made with purified fibrinogen, increased thrombin concentrations had faster rates of formation and lower maximum turbidity. In contrast, for clots made with plasma, increased thrombin concentrations had slower rates of formation and higher turbidity compared to clots made with lower thrombin concentrations.

Furthermore, we studied the coefficients of the generated equations from the multiple linear regressions using our data (Appendix A). In the case of diameter, we saw similar magnitudes for both coefficients for purified fibrinogen and plasma clots (10.89 vs. 59.89 for fibrinogen and −88.73 vs. −100.15 for thrombin). The sign of these coefficients describes the role of these concentrations on the diameter or other independent parameters. For example, in the case of diameters, the positive coefficients for fibrinogen concentration implies that increasing fibrinogen leads to larger diameters; contrastingly, the negative coefficient for thrombin concentration implies increasing thrombin leads to smaller diameters. All trends can be found in the Appendix A.

Next, we looked at the trends with plasma when made with TF rather than thrombin (Appendix A). These equations were generated using data from a fixed volume of TF and a fixed ratio of TF to fibrinogen to obtain a macroscale understanding of the role of fibrinogen and TF in these clots. Overall, the magnitudes of the coefficients for fibrinogen and TF concentrations are smaller compared to those for plasma clots made with thrombin.

## 4. Discussion

It is essential to study blood clot formation and clinically relevant fibrin network structures to pinpoint the role of selected blood proteins and infer patient risk from aberrancies in their concentration or biochemistry [15,28,50,51]. Here, we have systematically studied fibrin polymerization and the structure of clots made with purified fibrinogen and plasma across a range of physiological concentrations and compared results between different experimental techniques to identify overall trends.

Understanding the effect that changing fibrinogen concentration has on clot formation is important because heart disease and stroke have been linked with increased fibrinogen concentrations [48,51,52,53], while trauma-induced coagulopathy and bleeding have been linked to decreasing fibrinogen concentrations [54,55,56]. In studying the samples made from purified fibrinogen and samples made with plasma, many of the trends in alterations in clot and fiber structure were the same with changing fibrinogen concentration (Figure 1 and Figure 2, Table 1 and Appendix A).

Furthermore, increasing fibrinogen concentration caused a longer lag time for purified fibrinogen samples but a shorter lag time for the plasma samples. It would be expected that with increasing fibrinogen concentration, there would be a lower thrombin: fibrinogen ratio, leading to slower fibrinopeptide cleavage [57], and thus a longer lag phase. Furthermore, increased fibrinogen concentration provides more holes for the knobs on fibrin monomers to bind to, which would increase the lag phase. This explains the increased lag phase with increased fibrinogen concentration that is seen with purified fibrinogen. It has been found that gamma-globulin and fibronectin result in reduced lag phases [30], so it is possible that these plasma proteins are contributing to the decreased lag phase with increased fibrinogen concentration for the plasma samples, or it is possible that it is a difference in the reaction time of starting the measurements, since the difference was insignificant and much shorter than those seen with the purified fibrinogen samples. Future studies can further explore the role of other plasma proteins.

To isolate the role of fibrinogen concentration while excluding the confounding role of TF, we examined a fixed ratio of tissue factor to fibrinogen rather than a fixed concentration of tissue factor. With 1 mg/mL fibrinogen, the samples with a fixed ratio had thinner, shorter fibers and a smaller fluorescent area covered compared to samples with 2.7 mg/mL fibrinogen (Appendix A). They also had a longer lag phase and an increased rate of clot formation (Appendix A). Since a lower overall tissue factor leads to slower kinetics in the 1 mg/mL sample, the fibers had time to laterally aggregate more to create thicker fibers. Surprisingly, the pore size is similar between the fixed ratio and volume ratio samples; however, the increased fluorescent area covered is also likely due to the increased diameter. With more protofibrils incorporated into the lateral aggregation of the fibers, they did not grow as much longitudinally, as seen by comparing the fiber lengths in Figure 2 and Appendix A.

It is also important to understand the effect of changing thrombin concentration, as bleeding conditions, notably hemophilia, are characterized by insufficient levels of thrombin [58,59,60,61,62]. While purified fibrinogen had similar trends for clot structure and lag time (Figure 4 and Figure 5, Table 2 and Appendix A), reverse trends were found for the rate of clot formation and maximum turbidity (Table 2). An increased rate of clot formation for purified fibrinogen can be explained by fibrinopeptide cleavage being faster with increasing thrombin concentrations, resulting in faster protofibril formation [57], and the difference in the plasma samples’ rate of clot formation is not statistically significant with the changing thrombin concentrations. However, it could be affected by the presence of albumin in plasma, which has been shown to decrease the rate of clot formation [28,63,64,65]. This increasing maximum turbidity with increasing thrombin concentration for plasma was also found by Shah et. al. [30] in comparing thrombin concentrations of 0.25 and 1.5 U/mL, and it was hypothesized that it is because of antithrombins, resulting in less fibrinogen being converted to fibrin at low thrombin concentrations, with their effect decreasing as the thrombin concentration increases.

Altered fiber diameter has been linked to several diseases such as ST-elevation myocardial infarction, ischemic stroke [66,67], venous thromboembolism [14,68], diabetes [69], and COVID-19 [24,70,71]. Fibrin clot structure plays a role in fibrinolysis: changes to the fibrin network and individual fibers can make a patient more susceptible or resistant to lysis [15,24,39,72,73,74,75]. We investigated the fiber diameter using both SEM imaging and using turbidimetry for comparison with two different turbidimetric fitting approaches used. All three methods show an increase in the fiber diameter as the fibrinogen concentration is increased for both purified fibrinogen clots and for plasma clots. Similar to previous results comparing turbidimetry with super-resolution microscopy [44], we found good agreement between the diameter values obtained using the three methods for low fibrinogen concentrations with the Carr–Hermans approach providing the most similar diameter values to those obtained using SEM. However, much of the previous work did not explore concentrations higher than 1 mg/mL, whereas physiological concentrations range from 2 to 5 mg/mL and can be even higher in the presence of inflammation [47], COVID-19 [23,24,75], diabetes mellitus [22,69], and smoking [76,77,78]. It has been previously shown that as the fibrinogen concentration increases above 1 mg/mL, the corrected Yeromonahos approach is the most accurate in comparison to SEM imaging [35], and we find the same results for purified fibrinogen samples here. However, it appears from our results that as the fibrinogen concentration is increased, the variability between the reported diameters using the different methods increases, especially for the plasma samples. This makes sense since the turbidimetric approaches are assuming that the solutions are diluted and that the fibers are thin compared to the wavelengths being used for measurement. Therefore, as the fibrinogen concentration is increased and the fibers become thicker, the turbidimetric approaches become less reliable [37].

For the diameter values obtained using SEM and turbidimetry with changing thrombin concentration, all three methods show decreased diameter with increasing thrombin concentration for both purified fibrinogen and plasma, but the three methods overestimate relative to SEM. This could be due to the fibers shrinking during the drying steps required for SEM imaging, as the fibers were much thinner for the plasma samples than for the purified fibrinogen samples, which is the opposite of what has been previously reported [29,30]. This difference could also be due to prothrombin present in the plasma, which would mean that the thrombin concentrations are higher than expected, explaining why the plasma fibers are thinner than those for purified fibrinogen. The discrepancy between SEM and turbidimetry for the plasma samples could also be due to inaccuracy in the turbidimetric approaches, as described above. Since the diameters agree reasonably well between SEM and turbidimetry for purified fibrinogen, but not for plasma, it is also possible that the additional proteins present in plasma that bind to fibrin are leading to increased scattering, which is causing overestimates of the diameter values with turbidimetry.

Maximum turbidity is often used as a marker of clot structural changes, particularly diameter and density. Therefore, trends in how maximum turbidity changes can be used to predict these structural properties. We found that there was an increase in maximum turbidity for increasing fiber diameters for all conditions except plasma clots made with increasing thrombin, which may have had confounding influences, as described above. Similarly, there was an increase in maximum turbidity with increased percent fluorescent area for all conditions except clots made with purified fibrinogen at increasing thrombin concentrations. Interestingly, the linear fits and therefore linear relationships were stronger for samples that were formed with increasing fibrinogen concentrations. This implies there is more predictability for such comparisons.

After compiling structural and kinetic information, we were able to generate trends for clots from purified fibrinogen and plasma made with thrombin or plasma clots made with TF. These trends can aid in future studies to predict clot structure and polymerization characteristics to optimize experimental conditions. The multiple linear regressions were better fits for clots made with purified fibrinogen compared to those made with plasma. This suggests that a non-linear model may better fit plasma clots compared to the linear model. Nonetheless, these regressions provide a foundation to observe and predict trends in structure and kinetics of clots under specified conditions.

In this study, we have quantified the rates at which fibrin fiber and clot properties change as a function of coagulation molecule concentrations for both purified fibrinogen and plasma. This work provides a foundation to show how clot polymerization and structure change with altered fibrinogen and activation factor concentrations using a variety of experimental methods. The results presented here may also be useful to compare and interpret clotting from turbidity experiments with plasma samples from thrombotic patients, where the fibrinogen concentrations are often higher than controls, or in trauma patients or many dysfibrinogenemias, where there are low fibrinogen levels. It highlights not only how values change for varying clot concentrations but also how trends differ for clots created with purified fibrinogen and with plasma. Here, we focused on fibrinogen and clotting agent concentrations; future work could incorporate platelets to study glycoprotein V and platelet-generated thrombin that could alter clot structure. In addition, similar experiments could be performed with other factors in the coagulation cascade. For example, there is contradictory evidence on the role of FXIII on clot structure [79,80]; our combination of techniques could aid in the understanding of this role. Furthermore, recent studies have explored the role of mutant fibrinogens [26]; future studies could utilize our multi-technique approach to systematically analyze changes in clot structure due to mutant fibrinogen, genetic polymorphisms [81,82], and post-translational modifications of fibrinogen [83]. It would be expected that alterations in normal fibrinogen would impact the clot structure. Lastly, our clots were formed in the absence of flow, which resembles situations with the obstruction of blood flow; future work could study clot formation and structure with the presence of flow [84].

## 5. Conclusions

In this work, we studied clots made from both purified fibrinogen and plasma using an array of analytical techniques. We found that while changing fibrinogen and activation factor concentrations typically results in similar trends for changes in network and fiber structure for both purified fibrinogen and plasma clots, there were differences in the polymerization times and rates. Ultimately, our findings can help other researchers choose optimal conditions for their experiments and predict how results performed using purified fibrinogen will translate to physiological conditions where there are plasma proteins present. Furthermore, our results provide relevant parameters to develop or update mathematical models of clotting. In the long term, understanding the impact of the different experimental techniques and the varying clot conditions will aid researchers and clinicians alike in developing diagnostics and therapeutics.

## Figures and Tables

**Figure 1 biomolecules-14-00230-f001:**
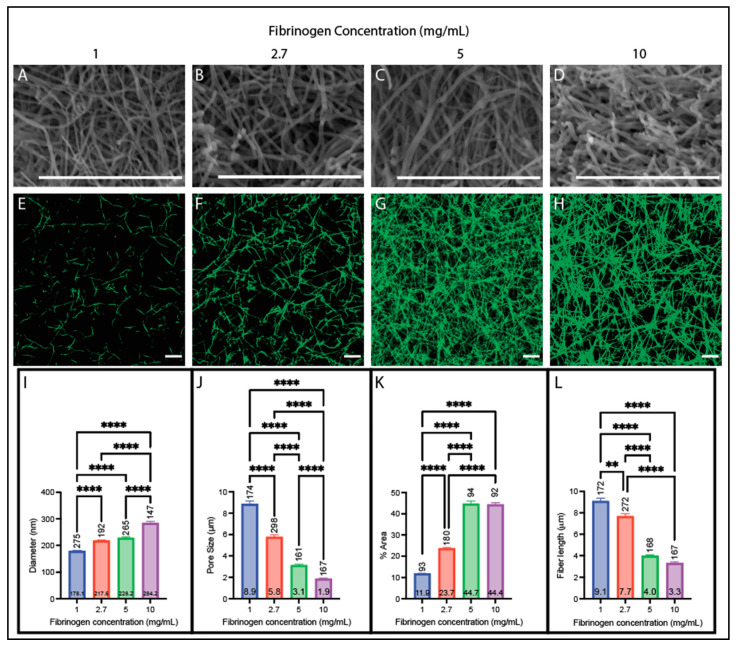
Structural analysis of clots made from purified fibrinogen with varying fibrinogen concentrations. (**A**–**D**) SEM images obtained on a Zeiss EVO10 (Zeiss, Oberkochen, Germany) at ~20,000× magnification on clots made with (**A**) 1 mg/mL, (**B**) 2.7 mg/mL, (**C**) 5 mg/mL, and (**D**) 10 mg/mL fibrinogen. (**E**–**H**) Confocal images obtained on a Zeiss LSM800 (Zeiss, Oberkochen, Germany) using a 63× oil immersion objective on clots made with (**E**) 1 mg/mL, (**F**) 2.7 mg/mL, (**G**) 5 mg/mL, and (**H**) 10 mg/mL fibrinogen. (**I**) Diameters acquired from SEM images. (**J**) Pore sizes acquired from confocal images. (**K**) Percent area covered obtained from confocal images. (**L**) Fiber lengths obtained from confocal images. (All clots contain 0.1 U/mL thrombin and 25 L-U/mL FXIIIa in a buffer of 150 mM NaCl, 20 mM HEPES, 5 mM CaCl_2_, pH 7.4; scale bars 10 µm;, ** *p* < 0.01, **** *p* < 0.0001.) Kruskal–Wallis with Dunn’s multiple comparison tests were used due to non-normal distributions present for each parameter. Horizontal number inside bar is the mean; vertical number above bar is sample size.

**Figure 2 biomolecules-14-00230-f002:**
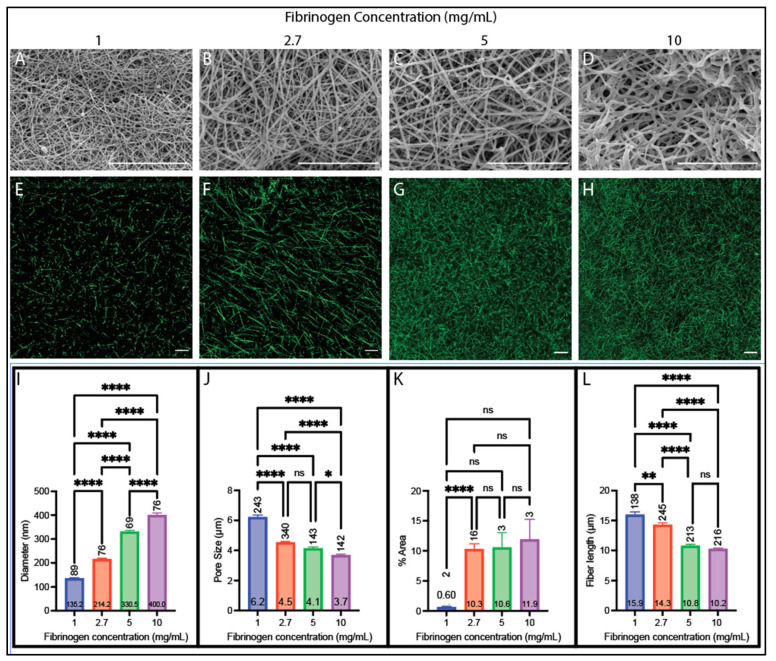
Plasma clot structural analysis with varying fibrinogen concentrations. (**A**–**D**) SEM images obtained on a Zeiss SIGMA (Zeiss, Oberkochen, Germany) at 20,000× magnification on clots made with (**A**) 1 mg/mL, (**B**) 2.7 mg/mL, (**C**) 5 mg/mL, and (**D**) 10 mg/mL fibrinogen. (**E**–**H**) Confocal images obtained on a Zeiss LSM800 using a 63× oil immersion objective on clots made with (**E**) 1 mg/mL (adjusted in FIJI for visualization), (**F**) 2.7 mg/mL (adjusted in FIJI for visualization), (**G**) 5 mg/mL, and (**H**) 10 mg/mL fibrinogen. (**I**) Diameters acquired from SEM images. (**J**) Pore sizes acquired from confocal images. (**K**) Percent area covered obtained from confocal images. (**L**) Fiber lengths obtained from confocal images. (All clots contain 1:80 volume ratio of tissue factor; scale bars 10 µm; ns not significant, * *p* < 0.05, ** *p* < 0.01, **** *p* < 0.0001.) Brown–Forsythe and Welch ANOVA tests and Dunnett multiple comparisons test (**I**,**K**), Kruskal–Wallis test and Dunn multiple comparison test (**J**,**L**). Horizontal number inside bar is the mean; vertical number above bar is sample size of individual measurements from 2 to 6 images.

**Figure 3 biomolecules-14-00230-f003:**
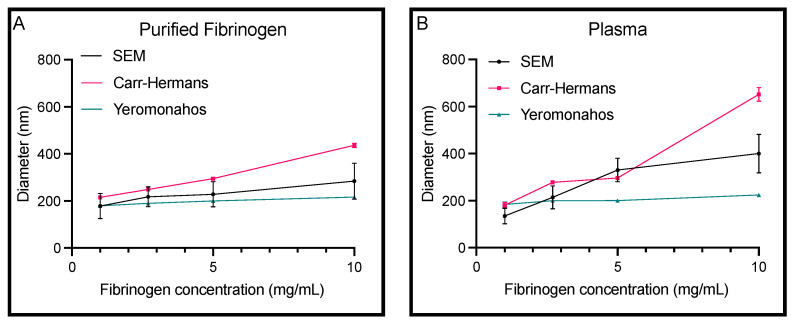
Diameter values obtained from the Carr–Hermans and Yeromonahos turbidimetric approaches as well as from SEM imaging for (**A**) purified fibrinogen and (**B**) plasma clots. Mean ± standard deviation displayed for scanning electron microscopy. Mean ± uncertainty due to approximation displayed for Carr–Hermans and Yeromonahos. *n* = 3.

**Figure 4 biomolecules-14-00230-f004:**
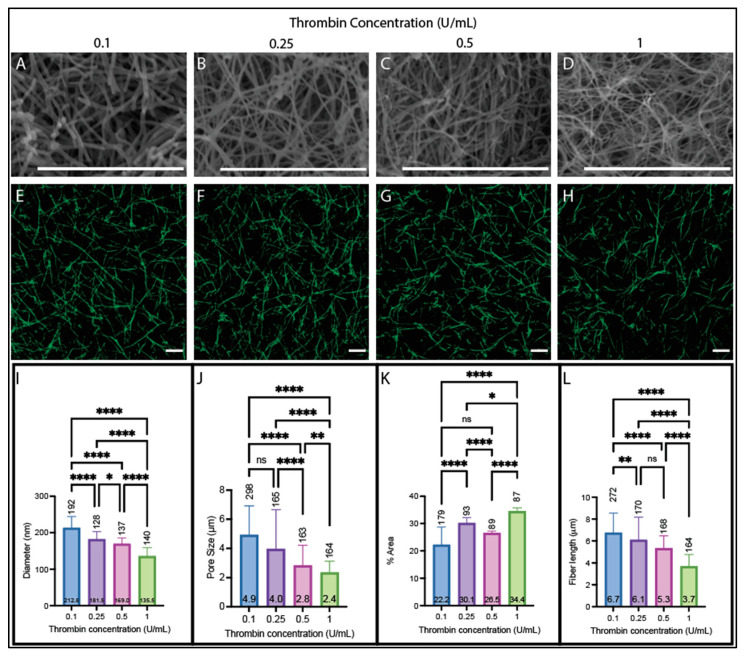
Structural analysis of clots from purified fibrinogen with varying thrombin concentrations (2.7 mg/mL fibrinogen). (**A**–**D**) SEM images obtained on a Zeiss EVO10 at ~20,000× magnification on clots made with (**A**) 0.1 U/mL, (**B**) 0.25 U/mL, (**C**) 0.5 U/mL, and (**D**) 1 U/mL thrombin. (**E**–**H**) Confocal images obtained on a Zeiss LSM800 using a 63x oil immersion objective on clots made with (**E**) 0.1 U/mL, (**F**) 0.25 U/mL, (**G**) 0.5 U/mL, and (**H**) 1 U/mL thrombin. (**I**) Diameters acquired from SEM images. (**J**) Pore sizes acquired from confocal images. (**K**) Percent fluorescent density obtained from confocal images. (**L**) Fiber lengths obtained from confocal images. (All clots contain 2.7 mg/mL fibrinogen and 25 L-U/mL FXIIIa in a buffer of 150 mM NaCl, 20 mM HEPES, 5 mM CaCl_2_, pH 7.4; scale bars 10 µm; ns not significant * *p* < 0.05, ** *p* < 0.01, **** *p* < 0.0001.) Kruskal–Wallis with Dunn’s multiple comparison tests were used to account for non-normal distributions present for each parameter. Horizontal number inside bar is the mean; vertical number above bar is sample size of individual measurements from at least 6 images.

**Figure 5 biomolecules-14-00230-f005:**
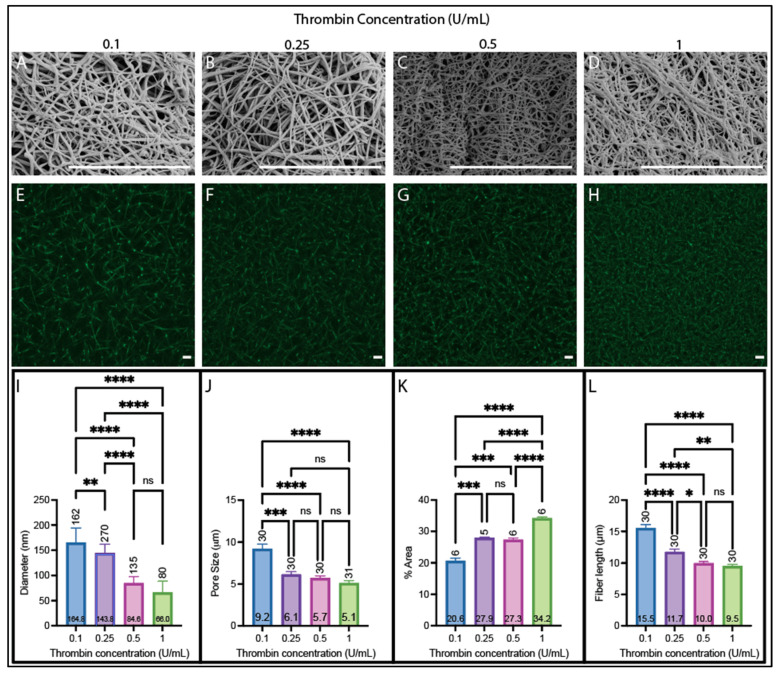
Plasma clot structural analysis with varying thrombin concentrations (2.9 mg/mL fibrinogen). (**A**–**D**) SEM images obtained on a Zeiss SIGMA at 20,000× magnification on clots made with (**A**) 0.1 U/mL, (**B**) 0.25 U/mL, (**C**) 0.5 U/mL, and (**D**) 1 U/mL thrombin. (**E**–**H**) Confocal images obtained on a Zeiss LSM800 using a 40x water immersion objective on clots made with (**E**) 0.1 U/mL, (**F**) 0.25 U/mL, (**G**) 0.5 U/mL, and (**H**) 1 U/mL thrombin. (**I**) Diameters acquired from SEM images. (**J**) Pore sizes acquired from confocal images. (**K**) Percent fluorescent density obtained from confocal images. (**L**) Fiber lengths obtained from confocal images. (Scale bars 10 µm; ns not significant, * *p* < 0.05, ** *p* < 0.01, *** *p* < 0.001, **** *p* < 0.0001.) Brown–Forsythe and Welch ANOVA tests, Dunnett multiple comparisons test (**I**–**L**). Horizontal number inside bar is the mean; vertical number above bar is sample size of individual measurements from at least 6 images.

**Figure 6 biomolecules-14-00230-f006:**
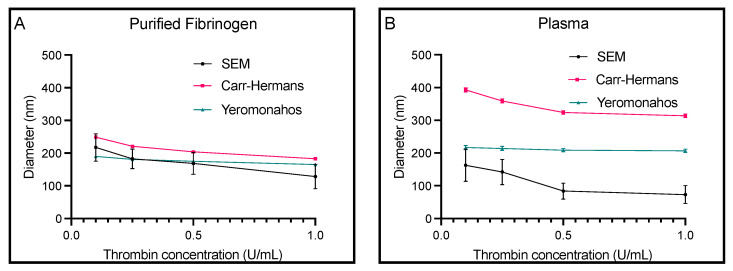
Diameters obtained from the Carr–Hermans and Yeromonahos turbidimetric approaches as well as from SEM imaging for (**A**) clots from purified fibrinogen and (**B**) plasma clots. Mean ± standard deviation displayed for scanning electron microscopy. Mean ± uncertainty due to approximation displayed for Carr–Hermans and Yeromonahos. *n* = 3.

**Table 1 biomolecules-14-00230-t001:** The lag phase, rate of clot formation, and max turbidity values for the clots from purified fibrinogen (left side) and plasma (right side) with fibrinogen concentrations of 1, 2.7, 5, and 10 mg/mL; ns not significant, *** *p* < 0.001, **** *p* < 0.0001. Dash (—) corresponds to lag time occurring before first reading. A standard deviation of 0 corresponds to the values of all biological replicates being the same. *n* = 3.

Parameter	Purified Fibrinogen					Plasma				
Fibrinogen Concentration (mg/mL)	1	2.7	5	10	*p*-value	1	2.7	5	10	*p*-value
Lag Time (s)	140 ± 0	155 ± 0	175 ± 5	210 ± 13	***	70 ± 36	45 ± 22	—	—	ns
Rate of Clot Formation (cm^−1^/s × 10^−3^)	3.4 ± 0.1	6.88 ± 0.06	8.2 ± 0.2	10.2 ± 0.4	****	11 ± 3	39 ± 3	50 ± 5	59 ± 2	****
Maximum Turbidity (cm^−1^)	1.85 ± 0.03	5.24 ± 0.06	7.85 ± 0.10	9.35 ± 0.21	****	0.87 ± 0.17	4.16 ± 0.10	7.23 ± 0.62	11.42 ± 0.19	****

**Table 2 biomolecules-14-00230-t002:** The lag phase, rate of clot formation, and max turbidity values for purified fibrinogen (left side) and plasma (right side) clots with thrombin concentrations of 0.1, 0.25, 0.5 and 1 U/mL; ns not significant, ** *p* < 0.01, *** *p* < 0.001, **** *p* < 0.0001. Dash (—) corresponds to lag time occurring before first reading. A standard deviation of 0 corresponds to all the values being the same. n/a *p*-value was due to the values within each concentration being the same (standard deviation of 0). *n* = 3.

Parameter	Purified Fibrinogen					Plasma				
Fibrinogen Concentration (mg/mL)	0.1	0.25	0.5	1	*p*-value	0.1	0.25	0.5	1	*p*-value
Lag Time (s)	105 ± 0	75 ± 0	30 ± 0	—	n/a	113 ± 25	24 ± 3	13 ± 3	—	****
Rate of Clot Formation (cm^−1^/s × 10^−3^)	9.9 ± 0.2	14.1 ± 0.5	24.5 ± 0.6	33.3 ± 1.5	****	2.3 ± 0.4	2.1 ± 0.4	1.6 ± 0.3	1.5 ± 0.2	ns
Maximum Turbidity (cm^−1^)	5.86 ± 0.14	5.20 ± 0.15	5.13 ± 0.09	4.47 ± 0.17	***	2.31 ± 0.27	3.02 ± 0.14	2.83 ± 0.17	3.51 ± 0.32	**

## Data Availability

The data presented in thus study are available on request from the corresponding author.

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
