# Peer review of "Comprehensive Analysis of the Role of Fibrinogen and Thrombin in Clot Formation and Structure for Plasma and Purified Fibrinogen"

_biomolecules, 2024, doi:10.3390/biom14020230_

Round 1

Reviewer 1 Report

Comments and Suggestions for Authors

Risman et al.'s study delves into the roles of fibrinogen and thrombin in clot formation. Their analysis focuses on the structural and polymerization characteristics of clots formed using purified fibrinogen or plasma under varying concentrations of fibrinogen and clotting agents. Notably, authors characterized fibrin polymerization and the structural properties of clots formed with purified fibrinogen and plasma. They employed various experimental techniques spanning a range of physiologically relevant fibrinogen concentrations to identify overarching trends in clot formation and structure. This well-written manuscript provides valuable insights into the intricate mechanisms of clot formation.

 However, I noticed some minor things that could be easily addressed.

There is a difference in sample size between the groups and across the comparisons. Could the authors please clarify the rationale for this unevenness?

For the human pooled plasma used in Figure 2 experiments, could the authors clarify: 1) How many individual subjects were used to create each pool? 2) Does the reported sample size represent independent replicate experiments performed using the same individual's pooled plasma?

Author Response

Risman et al.'s study delves into the roles of fibrinogen and thrombin in clot formation. Their analysis focuses on the structural and polymerization characteristics of clots formed using purified fibrinogen or plasma under varying concentrations of fibrinogen and clotting agents. Notably, authors characterized fibrin polymerization and the structural properties of clots formed with purified fibrinogen and plasma. They employed various experimental techniques spanning a range of physiologically relevant fibrinogen concentrations to identify overarching trends in clot formation and structure. This well-written manuscript provides valuable insights into the intricate mechanisms of clot formation.

However, I noticed some minor things that could be easily addressed.

Response: The authors thank the reviewer for recognizing the value of the paper and providing helpful feedback to ensure transparency.

  1. There is a difference in sample size between the groups and across the comparisons. Could the authors please clarify the rationale for this unevenness?

Response: The authors thank the reviewer for sharing this observation. Technical variability between experiments led to slight differences in sample size. For transparency, we provided the range of measurements in the methods.

  1. For the human pooled plasma used in Figure 2 experiments, could the authors clarify: 1) How many individual subjects were used to create each pool? 2) Does the reported sample size represent independent replicate experiments performed using the same individual's pooled plasma?

Response: The human pooled plasma was commercially purchased from a vendor. We requested the vendor to pool plasma from at least 25 healthy donors. 2) The independent replicates were performed using the same lot of pooled plasma. The following sentences were included in the manuscript to reflect this information, “Commercially available human-pooled plasma from more than 25 healthy donors was purchased from Cone Bioproducts (#5781). Individual replicates were conducted with the same lot of pooled plasma.”

Reviewer 2 Report

Comments and Suggestions for Authors

In this manuscript, Risman and colleagues provide a comprehensive analysis of fibrin clot formation using purified fibrinogen as well as plasma at different concentrations and using different techniques to assess fibrin structure. In my opinion, this work is clearly of importance to the field (as a kind of reference work), despite several similar studies (which – to the best of my knowledge are not as comprehensive). My only "major comment" is that it is not clear to me what the authors mean by sample size. Since it is quite high in most of the figures, I assume they mean the number of fibrin fibers analyzed, which would then be a misnomer. Since the authors are working with purified fibrinogen and/or pooled plasma, a classical biological n is unrealistic. Therefore, I would recommend reporting the number of technical replicates as "n" and noting the range of fibers on a separate note, e.g., n=3 technical replicates, each representing 15-67 fibers, in every figure legend. This would of course also affect the statistical test, as the fibers of each replicate would have to be averaged, so a true n=3 would be used for the statistical assessment.

Minor Comments:

1)     In my opinion, the title is a bit clumsy, and the authors are selling themselves short. How about "Comprehensive analysis of fibrin clot structure in plasma and purified fibrinogen in dependence of the fibrinogen concentration" or something similar?

2)     Two recent studies should be cited in the introduction: The very recent article of Hur et al., Blood 2024, 143:105-117 on fibrinogen mutants and their effects on thrombosis, as well as the work of Beck et al., Nat Cardiovasc Res 2023, 2:368-382 demonstrating that also soluble GPV affects fibrin formation (which might also be present in plasma).

3)     I appreciate the authors’ narrative way of introducing the methods and the transparency which experiments were performed in which institute. Nevertheless, the devices (electron microscopes, microplate readers) and their manufacturers should be stated in this section.

4)     Given the ever-increasing relevance of codes in image analysis and the problem of reproducibility, the authors should provide their python code used for multiple linear regression via Github, Zenodo or something similar.

5)     It would be even more reader-friendly, if the authors would indicate the fibrinogen concentrations in the representative images of Fig. 1, 2. 4 and 5.

6)     Figure 3 and 6 lack SD values (which links to my comment on the sample size above).

7)     The legends of the tables should also provide information on the sample size and their nature.

Reviewer 3 Report

Comments and Suggestions for Authors

1. Introduction is long and not focused. It can be shortened.  

2. Introduction- “Following vessel injury, levels of procoagulant factors are upregulated in the blood, 49 which can ultimately lead to fibrin polymerization and clot formation” I think ‘upregulated” is not a proper word here, Since procoagulant factors are already present during pathological conditions leading to thrombosis following vascular injury.

3. Some important considerations- These experiments are done in “in vitro conditions” with limited amount of blood constituents required for clot formation. This is significantly different from in vivo process -“under flow” that too under different flow conditions- high flow rate in arteries and low flow rate in veins that affect differently on clot generation and properties. This should be acknowledged and discussed in the MS.

4. The authors should also discuss the potential influence of genetic polymorphisms and post translational modifications of of fibrinogen on clot structure.

5. Results and discussion sections are too long.

6. It looks like concentrations of fibrinogen, TF and thrombin are not physiological concentrations. 1-2.7mg/ml fibrinogen may be too low and may reflect bleeding side. Please clarify. For example, the fibrinogen concentration in high risk patients with CVD is in the range of 600-800mg/dl and much more in patients with COVID-19 (our own experience).  Therefore, current experiment ( in vitro assay and lower conc of  fibrinogen and thrombin) may not be applicable for pathological conditions such as heart attack or stroke.

7. Moreover, these experiments were done in the absence of activated platelets that play a major role in thrombin generation and clot formation.

8. The authors should discuss the above limitations of this experiment.

9. The authors should check the reference section carefully. They are not in an uniform format and some references are incomplete.

Comments on the Quality of English Language

MS is too long and can be streamlined and focused. 

Author Response

Reviewer 3:

  1. Introduction is long and not focused. It can be shortened. 

Response: The introduction was reorganized and shortened.

  1. Introduction- “Following vessel injury, levels of procoagulant factors are upregulated in the blood, 49 which can ultimately lead to fibrin polymerization and clot formation” I think ‘upregulated” is not a proper word here, Since procoagulant factors are already present during pathological conditions leading to thrombosis following vascular injury.

Response: We have changed the sentence to say, “ “Following vessel injury, levels of procoagulant factors are upregulated activated in the blood, which can ultimately lead to fibrin polymerization and clot formation”

  1. Some important considerations- These experiments are done in “in vitro conditions” with limited amount of blood constituents required for clot formation. This is significantly different from in vivo process -“under flow” that too under different flow conditions- high flow rate in arteries and low flow rate in veins that affect differently on clot generation and properties. This should be acknowledged and discussed in the MS.

Response: The reviewers acknowledge the absence of flow as a limitation. Standard coagulation studies with limited components, as we have done, have been performed for decades, but even in those studies, there is a gap in comparing results between different techniques, which we are trying to fill. We agree that it would be nice to use these techniques to study more comprehensive systems in the future. We have included this in the Discussion section by saying, “Lastly, our clots were formed in the absence of flow, which resembles situations with the obstruction of blood flow; future work could study clot formation and structure with the presence of flow.”

  1. The authors should also discuss the potential influence of genetic polymorphisms and post translational modifications of fibrinogen on clot structure.

Response: The authors thank the reviewer for this suggestion as this is an important advancement in the field. Our studies provide a necessary foundation for all such future studies. We have included this as a limitation and next step in the Discussion by saying, “Furthermore, recent studies have explored the role of mutant fibrinogen [23]; future studies could utilize our multi-technique approach to systematically analyze changes in clot structure due to mutant fibrinogen, genetic polymorphisms [71,72], and post-translational modifications of fibrinogen  [73]. It would be expected that such alterations in native fibrinogen would impact the clot structure.”

  1. Results and discussion sections are too long.

Response: The results and discussion sections were edited for length. In particular, we removed repetition between sections.

  1. It looks like concentrations of fibrinogen, TF and thrombin are not physiological concentrations. 1-2.7mg/ml fibrinogen may be too low and may reflect bleeding side. Please clarify. For example, the fibrinogen concentration in high risk patients with CVD is in the range of 600-800mg/dl and much more in patients with COVID-19 (our own experience).  Therefore, current experiment ( in vitro assay and lower conc of  fibrinogen and thrombin) may not be applicable for pathological conditions such as heart attack or stroke.

Response: Our experiments covered concentrations from 1-10 mg/mL; this range was chosen to reflect both low or bleeding (1 mg/ml), normal or healthy (2.7 mg/mL) and high or hypercoagulable conditions (5 and 10 mg/mL). In the cases of TF and thrombin, concentrations were chosen based on those used in similar standard clotting assays (J Thromb Haemost. 2018 May;16(5):1007-1012.).

  1. Moreover, these experiments were done in the absence of activated platelets that play a major role in thrombin generation and clot formation.

Response: The authors acknowledge the importance of platelets to the coagulation cascade and agree it is valuable to include these in future similar experiments. In the current study, we wanted to focus on isolating fibrinogen and clotting factors. We followed standard assays in the literature for many years that use such simplified components. This point has been added to the Discussion by saying, “Here, we isolated the effects of fibrinogen and clotting agents; future work could incorporate platelets to study glycoprotein V and platelet-generated thrombin that could alter clot structure.”

  1. The authors should discuss the above limitations of this experiment.

Response: The authors have stated the limitations in the Discussion. Individual responses can be seen above.

  1. The authors should check the reference section carefully. They are not in an uniform format and some references are incomplete.

Response: The authors thank the reviewer for this observation. We have carefully reviewed the references and adjusted the citations as needed. Furthermore, we changed the style format to MDPI-Chicago as suggested by Biomolecules.

Round 2

Reviewer 3 Report

Comments and Suggestions for Authors

1. The authors discuss in detail about influence of thrombin concentrations. It is important to know that thrombin generation is a dynamic process during clot formation, particularly pathologic clot formation at the site of vascular injury and it also influences the evolving clot structure- more dense platelet rich clot  at the bottom at the site of injury (primary hemostasis) with vey low levels (<1nM)of thrombin followed by generation of 100 - 200 times higher concentrations of thrombin on the surface of platelets and  subsequent generation of thrombus with platelets and more fibrin formation at the periphery (secondary hemostasis) (Example. Dr. KG Mann’s work).  In this line, how study of one concentration at a time during clot formation as used in the current study will provide information on in vivo pathological clot formation?
2. Another interesting observation regarding TF – difference between TF exposure during hemostasis. Significant amount of TF exposure influencing high levels of thrombin generation at the outer layer of blood vessel during hemostasis as compared to low levels of TF exposed initially at the site of pathological injury in the presence of endothelial injury and subsequent activation coagulation system in the presence of activated platelets (as discussed in the PMID: 34353538). How this process influence clot structure during hemostasis and pathological thrombus formation?

3. Will the analysis of different fibrinopeptides released during clot formation help us to understand the fibrin polymerization and clot structure?

4. Finally, FXIII significantly influences fibrinogen polymerization and 3D structure of the fibrin network. Why the authors did not study and discuss this in the manuscript.

5. Finally, the authors should be congratulated for their comprehensive initial experiments in understanding the generation and structure of clot.

Comments on the Quality of English Language

None
